# Refuge and Resistance: Theater with Kurds and Yezidi Survivors of ISIS

Ellen Wendy Kaplan 

Theatre Department, Smith College, Northampton, MA 01063, USA; ekaplan@smith.edu

**Abstract:** This essay looks at ongoing efforts to revitalize arts and culture among the Yezidi and broader Iraqi Kurdish communities. The Yezidi are survivors of the 2014 genocide perpetrated by the Islamic State (ISIS, also known by its Arabic acronym Da'esh) which resulted in mass killing, captivity and expulsion from their ancestral homeland of Mt. Sinjar in northern Iraq. They are part of the Kurdish people, who have engaged in centuries of struggle to protect their cultural and political identity, establish autonomy and ensure their security in the broader Middle East. After a brief overview of the Yezidi genocide and its aftermath, we trace some theatrical efforts in the 20–21st century and look at two embryonic theater initiatives in Iraqi Kurdistan. The description of cultural projects at Springs of Hope Foundation (Shariya Camp) is followed by personal reflection and analysis of the aims, uses and challenges of Applied Theater. This 'umbrella term' refers to a process that uses a theatrical tool-kit in non-theater contexts. The aesthetic, ethical and political challenges inherent in this work are considered: the essay explores questions of ethical care and the implications and pitfalls of working with vulnerable and displaced populations, issues of representation, and creating spaces for healing and expression through participatory theater. Finally, we discuss a new initiative in Iraqi Kurdistan that seeks to address ethnic and political fissures through theater. The essay culminates with a consideration of belonging and re-imagining home.

**Keywords:** Yezidi; Kurds; Applied Theater; Springs of Hope; ME-T

"If we weren't strong, we wouldn't be here". Gawre, resident of Shariya Camp.

## 1. Springs of Hope, Shariya Camp, Iraq

In the blazing heat of August, seven young Yezidi women in simple chiffon robes seem to float into a plaza filled with spectators. As they enter, they are preceded by dozens of young men and women in traditional Kurdish dress shimmer in the bright sun: women in long flared skirts and sequined vests, a few in hijab, and men wearing round multi-colored hats, loose trousers and cummerbunds. They are clapping and playing the *daf*–a traditional Kurdish drum in resounding rhythm. This is Kurdistan, a semi-autonomous region of northern Iraq; Yezidi residents of Shariya Camp for Displaced Persons are commemorating the 7th anniversary of the 2014 genocide, when ISIS murdered, enslaved, and expelled thousands of Yezidi people from their ancestral lands.

Somber and serene, the women enter a plaza filled with spectators and streaming with light. Slowly, they encircle a small fountain. Their robes are shaded from black to slate gray, to light sea blue, to pure white. Each carries a small blue stone, which represents the heavy burdens of memory; one by one, solemnly, they kiss their stone, take a few steps to toss it away, and return to the circle. The girl in white, who led the seven girls into the plaza, crowns each one with a tiara. They join hands and lift their arms to the sky, celebrating themselves, their hopes for the future, and their dreams.

Since 2015, Springs of Hope Foundation for Relief and Development in Iraq (SOHF) under the leadership of Lisa Miara (Founder, President) and Dr. Alo Saeed (CEO) has served hundreds of women, teens and children who reside in Shariya Camp and the pre-existing Yezidi village of Shariya[1] in Iraqi Kurdistan, near the small city of Duhok.[2] The

camp houses some 20,000 Yezidi internally displaced people (IDP),[3] most of whom fled their homes on Mt. Sinjar (part of the Shengal region, alternately Shingal) in 2014. SOHF sits directly adjacent to the camp, occupying a complex of buildings that house classrooms, a health center, a large playground and sports area, and a newly built stable for three horses and Zamir,[4] a just-born foal. In addition to educational programming, vocational training, and group therapy, SOHF serves as a social hub, fostering community and connection for people who have spent eight years living in tents.

## 2. Yezidi History

Until the August 2014 massacre, the Yezidi were little known outside of Iraq. They are an ethno-religious group indigenous to Kurdistan—a region that includes parts of Iraq, Syria, Turkey, and Iran. The Yezidi community is one of the oldest in the Middle East, with roots in the ancient Medean peoples of Mesopotamia. Most Yezidi speak Kurmanji, a Kurdish dialect, though policies of Arabization throughout Kurdistan have meant loss of language, land and livelihood for minority communities, including Iraqi Kurds (the majority of whom identify as Sunni Muslim, or secular), Assyrian, Chaldean, and other minority communities in northern Iraq.

There are approximately one million Yezidi, more than half in diasporic communities in the US, UK, Germany and Australia.[5,6,7] Whether or not Yezidi are Kurds is "a matter of dispute among scholars, Kurds and Yezidi themselves, as to whether they are ethnically Kurds or form a distinct ethnic group". ([Khattar 2022](#)). Yezidi say they are the "original Kurds", although some identify as a subset of the Kurdish people, while others identify as a separate ethno-religious group. The religion has pre-Zoroastrian roots, is orally transmitted, and is syncretic, incorporating aspects of Abrahamic and non-Abrahamic faiths.[8] At Lalish, the holy compound deep in the mountains of Nineveh Province (presumed to be 4500 years old), a doorway to the main temple features the image of the snake that, we were told by one of the caretakers at the temple, saved Noah's ark from sinking by inserting itself into a hole in the boat. Other Yezidi customs, including baptism with consecrated water and anointing with oil, have parallels in Christianity. But, Muslims do not consider Yezidi to be '*ahl al-kitab*', or 'people of the book' mentioned in the Quran.[9]

Yezidis say they have experienced 73 *fermans*, or pogroms, over their long history. With the introduction of Islam to the region in the 7th–8th centuries, Muslim clerics charged them with heresy stemming from the mistaken belief that Yezidi people worship the devil. (God's representative on earth, the Peacock Angel Melek Taus, is also called Shaytan, incorrectly translated in Arabic as Satan). The Ottoman empire conducted campaigns against them beginning in the 17th century through its collapse in 1922; in Republican Iraq (post 1958), Pan-Arab nationalism made targets of Yezidis and Muslim Kurds. After the 1974 Kurdish uprising, and again in the aftermath of the Iran–Iraq war (1980–1988), Saddam Hussein confiscated Yezidi land; in the 1990s, pan-Islamism inflamed biases toward religious minorities. After Saddam's death in 2006, Al-Qaeda murdered hundreds of Yezidi as it swept across Syria and Iraq. On 3 August 2014, ISIS massacred thousands on Mt. Sinjar: Yezidi men were murdered, women were kidnapped and sold into sexual slavery, small girls were used as domestic servants, boys were forcibly converted to Islam and forced to fight as jihadis.[10] Dispossessed, their homes and families destroyed, hundreds of thousands fled. Those who escaped were housed in camps throughout KRI (Kurdish Region of Iraq).[11] Mass graves are still being uncovered on the mountain, and the threat of renewed violence is increasing.[12] The explicit intention has been permanent demographic change and territorial gain accomplished through the extermination, forced conversion, expulsion, and permanent dispossession of the Yezidi people.

In 2019, I spoke with Baba Chowith, a spiritual leader (*pir*) who resides at Lalish Temple, the holiest site of the Yezidi. "We are tired", he said. "We are broken. There is nothing left". He shared his thoughts about the dilemma facing the Yezidi: the young women captured on Mt. Sinjar were sold, resold, repeatedly raped, forced to convert. They were taught to hate their families; those that returned could no longer speak their own

language. "We welcome them back, we forgive, if we did not, there'd be no more Yezidi".[13] Although the women are re-integrated, their children, born of non-Yezidi fathers, were not welcomed. The community still wrestles with this.

Conditions on the mountain continue to deteriorate; homes are turned to rubble, basic infrastructure is nonfunctional, and by May 2022, a tentative agreement with Yezidi forces has broken down, Turkish incursions near Mt. Sinjar, and the failed agreement between Yezidi protection forces and Turkish and Iraqi government forces is causing continued instability.

### 3. Bringing Theater to SOHF

In 2019, I intended to spend several weeks in Rojava, a Kurdish-led socio/political experiment in northeast Syria.[14] In the previous year, I interviewed several dozen Kurdish activists, artists, emigres and refugees in the US and Germany who had lived in or had relatives in Rojava.[15] But as fighting against Isis raged, I was unable to procure permission to cross the border from Iraq into Syria. I remained in Iraq, and with help from Kurdish friends and local service organizations, I visited several IDP camps (Essayin, Gawilan, Khanke, and Shariya) in the Kurdish Region of Iraq and spoke with over a dozen severely traumatized women. All had fled their homes, most were still hoping to ransom or rescue their children in "Raqaa"[16] and all had lost multiple family members in the 2014 attacks. At the invitation of Kurdish–Iranian actor Mozarref Sheife,[17] I wrote *Testimonies*,[18] a play that incorporates verbatim accounts of the genocide and aftermath as recounted by Yezidi women. The stories they shared were chilling, and they reflected both grief and resilience in the face of unspeakable horror.

In May 2022, I returned to Iraq to work with Yezidi IDPs whom the Springs of Hope Foundation (SOHF) serves. Most of the students at SOHF fled Daesh in 2014; some were born in the camp; many are orphans or are sole survivors of their tribe. They did not have a childhood. They have few memories other than of devastation. After years of servitude, they speak only Arabic; their names have been changed. A large percentage are illiterate. They have been living in tents for the last eight years.

Since its inception in 2015, SOHF has served Iraqi Arabs escaping the ISIS invasion of Mosul (June 2014), Syrian Kurds fleeing the Assad regime during the civil war, Yezidi from Shingal, and other minority communities in Kurdistan. A broad set of educational programs and psychosocial support work includes classes in sewing, music, art, computer literacy; coding and programming, and ESL; mental health services, including play therapy, psychotherapy, and equine-assisted therapy. In a post-genocide society, the needs are enormous and resources scarce; on the ground organizations such as SOHF offer safety, care, respect, social support, and dignity.

My work at SOHF was to offer a theater program to create supportive, playful spaces for creative expression. As a director and writer, I make "traditional" theater; I perform in and write conventional plays. I also work 'outside the building' with diverse groups who do not know, do not like—or cannot afford—mainstream theater. I have worked extensively with inner city, rural and special needs students across Pennsylvania; theater and literacy with teens and adult pre-GED students; with elders; at risk-youth, adjudicated teens, and in prisons.

The strategies and practices I employ derive from my 40 years of training and teaching acting, honed in theater work across borders, in conflict zones, and with under-resourced communities. This work is a type of Applied Theater, which is defined as a "broad set of theatrical practices, involving community participation, taking place outside of traditional theatre spaces", (Prentki and Preston 2009, p. 9)[19] responsive to people and communities, honoring particularity of story, history and culture, in service of social transformation. "The boundaries between actors and spectators are purposefully blurred as all participants are involved as active **theatre** makers" (O'Connor & O'Connor, 1).

The work is participatory, inclusive and non-didactic; it is defined not by message (what it says) but who it is by and for. In part, it is influenced by the path-breaking

work of Augusto Boal[20], whose Image Theater and Forum Theater offer a template for addressing community concerns; participants draw from a well of creativity and curiosity that makes each exploration exciting and fresh. The primary goals are inclusion, process, and becoming: in Levinasian terms; this pays more attention to the Saying, and less to the Said, as we examine below.[21]

Work with vulnerable communities requires a focus on the interests, concerns and priorities of the participants and respecting their agency, autonomy and dignity. I am aware of the limits of what I can offer. I lead and offer structure, but as Guglielmo Schinina asserts: "one first asks the community what its priorities are in order to understand its resources. Then one must adapt one's competencies to meet those needs". (Schinina in Balfour 2012, pp. 169–75).

The goal is to create positive healing spaces, as Yuko Kurahashi's discussion of Ping Chong's Children of War project demonstrates (in Balfour 2012, p. 254), not as therapy but as psychosocial support. Gentleness is key, as is humility. Part of what is required is, as Carol Gilligan writes, a radical listening[22], a deep, relational listening, predicated on an ethics of care, which may build connections with participants-as-creators. As a facilitator, I am aware of the trepidation with which many in this welcoming but vulnerable community approach creative play; I am a stranger here, and we need time to build trust. We work slowly, with the assurance that everyone participates only to the extent they desire, and no more. The goal, which we articulate at the beginning of each workshop, is to cultivate "self-esteem, self-confidence and self-expression" (Rostami 2019, p. 164).

Our sessions are tailored to meet the needs and interests of each group, with people of different life histories, levels of trauma and personal damage. (We do not engage with or address past trauma, but memories emerge unbidden, both in and out of workshops and classes: a pregnant woman, already a mother, suddenly begins to cry, recalling her father, who has joined ISIS; another, Gawre (quoted in the epigraph) recounts that of her five children, only three are with her, the other two are still missing in "Raqaa"; a young woman has a violent, psychotic break and has to be taken to hospital, but there are few resources for needs assessment, psychiatric services or medications in the camp. The young women in the workshop needed time to process this; we sat in silence, held and massaged hands, hugged, and little by little, they began to speak about their care for each other; they made drawings, and one girl drew a symbol of infinity—life goes on, she says. By the end of the hour and a half, we were interviewing each other on our cell phones, entertaining each other and laughing out loud.)

Cultural competency is crucial, though even with deep study and familiarity with Yezidi history and customs, and many Yezidi friends and acquaintances, I am still a stranger. The students assured me that the simple presence of an outsider is gratifying: they fear that they are invisible, unseen by history. Throughout my time at SOFH, I relied on the teaching and administrative staff to help negotiate cultural differences. Sahla Eanes, a Yezidi woman from Shariya village who directs the Hope Center at the camp, was an invaluable translator and partner as we played, explored physical expression, imaginative story building, and staged stories old and new. Eanes explains what to keep in mind: [23]

> Everything is changing. Here in the village, you cannot better yourself, there is no opportunity. But in the city, perhaps you can. Some years ago, Shariya was like a village, now it is like a city, and Duhok is near here—so the culture changes. If you see social media you see new things, you want new things. You have new dreams. Things are lost, but if something is not working then you need to change. If something is beautiful but useless, you have to change. Before, people don't go to school. There was no communication with the world. Now parents say, go to school. Now they encourage the kids to learn. But even now, our culture won't accept all the changes.

> Few can read. Especially women are illiterate. But they see a different life for their daughters. They don't know what will happen in the future. But—they can't go

back. Shingal—the impact of Arab culture there is too strong. They dress like Arabs. They have Arab names, they were neighbors. But still they are Kurds. Yezidi.

With the younger children, we introduced games that engaged their physical energy and delight in play to build intra-group cooperation and trust. Gleefully, they soared around the space, played with imaginary balls of fire, acted out original stories, and became superheroes with self-selected powers.

With older groups, including a therapy group led by a psychologist who invited me to offer a session, we began with breath, sensory awareness, mindfulness (I used Jon Kabat-Zinn's 'raisin' exercise, in which we slowly and with full awareness, smell, touch, taste and eat a single raisin), and being present in the body. It is not easy for traumatized people, whose senses have registered excruciating pain, to focus on the moment; we worked slowly, patiently, to the limits of each person's ability to concentrate, and then discussed the difficulty and power of this work. The Yezidi have a strong connection to nature, and cultivating this led to work on centering the self, breathing and relaxation, and imaginative visualization, which was expressed through drawing, poetry, simple improvisations and wide-ranging discussion.

Building trust and emphasizing cooperation and support were primary goals; cultivating imagination and responding physically to imaginary stimuli were important guideposts. Exploring the expressive potential of the body is a key element of the work, which is modulated by the comfort and ease of each member of the group at any given time.[24] We explored story building in depth; strategies include physicalizing responses (solo and in groups) to evocative words the group proposed: birth, growth, joy, hope, vanquishing fear; to imagining the story of photos of Kurdistan (mountains, animals, flowers, roads); drawing 'thumbprint' characters, imagining oneself to be a leaf (what kind? On what tree? Where? Who is there with you? What happens to the leaf? How does the story end?). We found stories in abstract paintings; in photos of something beautiful students took with their phones; we built stories from Kurdish proverbs, and from single lines of Rumi's sublime poetry. We staged classic Kurdish stories, including *Kawa the Blacksmith*, which is a foundational tale of Kurdish nationhood.[25]

Collectively, we developed original fairy tales, using a basic quest structure (a girl/boy walks into the woods to find something important; gets lost; meets an animal helper; finds a magic object; encounters an obstacle; with help from the animal and the object, the hero/ine overcomes the obstacle and gets what s/he wants, to a happy end. We shaped these stories gradually, encouraging each contribution, discussing options, analyzing and then acting out the stories as a group.

A cornerstone of the process was to delve into salient issues that the older students (ages 14 to 26) wanted to explore. We used a structure of three actions, beginning with a middle 'frame', that shows a frozen image of an identifiable problem. We then stage a 'before' frame (roots of the problem), and an 'after' frame—a potential solution or outcome. Drawing from Augusto Boal's work, we asked the other members of the group to imagine what the frozen statues might be saying, and then, the statues themselves spoke what they were thinking. Using mimesis (a drama therapy technique in which the group asks questions and the actor responds in character), each actor develops their characters and then creates short scenes about the issues that emerged, including bullying, drinking, and lack of productive work. Finally, we explored gender differences and limitations: we physicalized 'portraits' of men and women, looking at how we perceive each other, how gender determines education and job opportunities, and how things are changing as the younger generations (many of whom have no direct memories of their culture) are exposed to film, TV and social media.

Many of the participants value their experience with SOHF; they recognize their transformation over time and look to the future. One by one, they show how they have changed since they first arrived. From closed, isolated, collapsed bodies, they opened their torsos, faced outward, and gradually reached out to touch others. Strength, confidence,

and friendship were gradually achieved. "Before I had nothing, here I am a princess", says a fourteen-year-old girl who had fled Mt. Sinjar when she was six years old.

After the physical transformations from past to present, we considered generational change. Two empty chairs sat in front of the group, each an imagined space which holds the roots and branches of their families. First, they were to call into being their great-great-grandmother (or grandfather, as they wished). One by one, they approached the empty chair, addressed this ancestor with his/her (imagined) name, asked for advice, and received a gift. Then, we repeated the process, imagining a great-grandchild; they would name her, give a blessing, give advice, give a gift. This led to discussions of heritage, generational change, of what gets passed down and what can be carried forward. The past and the future are connected through the now.

We made portraits of the camp, of Kurdistan, of the world now and how it might look in 2050; we made a 'newscast' from the future. The students were eager and curious; some spoke a lot, others had very constrained ideas about the 'world out there' and the future we all face. Later, they finished the sentence, "When I look up I see . . . ", which led to mythic stories about other worlds, to talk about space and to looking at recent photos of the black holes at the center of the Milky Way. We discussed possible universes, as described in Yezidi lore, and in modern science. Both versions inspired awe.

Some imagined easily, others had trouble fantasizing or speaking about what they felt they did not know. But by the end of our time together, everyone engaged (whether still a bit shy or lustily involved), we made theater through exercises and strategies such as self-interviews, duologues, storytelling, role playing and improvisation, to create a joyful noise. Like music, one camp resident said, theater "is full of hope".

The most moving event I attend at SOHF is "Art in the Park". Dozens of small children are given easels and paints or markers and paper and sit in a wide circle. In a postage-stamp size local park with stunted trees and little greenery, they paint and draw while people from Shariya Village join or look on. The kids are concentrated, happy, proud; having an audience brings them to life, and they paint with abandon. Older boys from the camp sing and play rousing Kurdish music on *daf* and guitar. After the kids finish their artwork, and they receive free balloons and juice drinks, they gather in a thick knot of excited kids sitting in front of the band. The musicians jump into an energetic, Kurdish call-and-response song, which everyone knows. There are gales of laughter and energetic choruses from the kids. The musicians seem to be watering the children, nurturing, nourishing them, strengthening these kids with their music. The children are visible; they are loud and joyful; they are seen and heard.

## 4. Writing and (Re)presenting Stories on Behalf of Others

Doing theater work *with* vulnerable groups and writing plays *about* them are clearly different processes, with aims that may be aligned (or not), but they have different outcomes. But significant aesthetic, political and ethical challenges arise when we bring the work to a wider (non-refugee/non-IDP) audience: Anna Street notes three dilemmas artists face when engaging in this work: speaking on behalf of the other; aestheticizing trauma; and reinforcing oppositions (Street 2021, p. 2). Texts written by outsiders can generate problematic, reductive simplifications, in which the representations and performance of victimhood is presented as a spectacle, compelling empathy through, a "quagmire of personal narrative and victimhood discourse" (Balfour 2012, p. 219).

Many of my plays[26] integrate verbatim testimony and documentary evidence to consider individual trauma as it is linked to social and political dynamics. Verbatim plays are "one of the most common forms of theatre in this field reflecting an urge to maximize the truth claims for the work and generate a sense of authority for the stories presented" (Jeffers 2011, p. 80). Typically, this work is "not so much concerned . . . with ideological as with affective transformations in their audiences" (Burvill 2008, p. 234). But, when these audiences are comprised of the relatively privileged, there is an implicit (or explicit) call

to action; audiences are asked to assume a level of responsibility for official policies that create or contribute to the problem.[27]

Traumatized communities need to be heard and seen. But who tells the story? The aesthetic challenges are imbricated into the ethical: when we make theater on behalf of others who are not "here": whose voices, who embodies the stories? Who performs on stage, even as we agree that performance turns any space into a stage? Many people do not want to perform in public, others do not have the technical skills or training to effectively perform, and at times "performing" a traumatic event can re-trigger. In South Africa, for example, many witnesses for the Truth and Reconciliation Commissions requested that their testimony be performed by actors, who brought the stories to villages and towns around the country.[28] The victims could not, would not, repeat their trauma, but they wanted their stories told. "Telling", as James Thompson says, is not a neutral act (Thompson 2005, p. 25). Moreover, stories are not "owned", and giving dramatic structure to painful experience can be deeply meaningful.

We organize our experiences by telling, retelling, "mastering" our stories, making them our own. Narratives shape our mental landscapes, creating fluid and contested identities, which in turn are shaped by our personal and communal identities. Joanna Higgs, in her discussion of child soldiers among the FARC, brings in Husserl's concept of 'life worlds' which she defines as "a world of meanings and understandings that are socially, culturally and historically situated". (Higgs 2020, p. 79). Stories, shared and shaped collectively, scaffold the life worlds we inhabit.

In considering how to represent marginalized or vulnerable people, ethical concerns are paramount. There is the risk of exacerbating stereotypes of suffering and victimization, or there is a compensatory antidote that over-emphasizes assimilation and notions that refugees should be 'just like us.' There are questions about agency and artistic control: Should refugees be represented or represent themselves? Should they be "granted" agency, voice and visibility in order to increase their chance of being "heard" and "understood" by the audience and by the authorities (Blomfield and Lenette 2018, p. 325)? "Representation is a complex business", says cultural theorist Stuart Hall, especially when dealing with "people and places which are significantly different from us" (Hall 1997a, pp. 225–26, cited in Petersen 2021).

As outsiders, we are only a temporary part of the communities we visit, and simply by our presence, we are in a position of relative power.[29] As a Roma father in Bulgaria whom I interviewed in 1996 told me, "You can come here. We can't go there. Please tell our story". When we write and produce texts, we are helping to make our participant-colleagues visible. The subjects of our inquiry are traditionally without a public voice. Too often, they are merely "symbols of suffering" (Hamburger et al. 2021, p. 56).

Helene Cixous, discussing her collaboration with director Ariane Mnouchkine on *Le Dernier Caravansérail*[30], says the author has to disappear so that the other may appear (Cixous quoted in Jeffers 2011, pp. 73–74). She asks, should the author be there at all? The author may be present as a framing device, providing information and context, or as an avatar of empathy, a touchstone for identification. The question of representation, however, remains. Whose words are extracted, who voices those words, and to what end(s)?

Trauma numbs, silences, robs us of voice. Elaine Scarry suggests that in the moment of the infliction of pain, language disintegrates, and it is only after that moment that it can be recreated and shared, stating that "physical pain has no voice, but when it at last finds a voice, it begins to tell a story" (Scarry 1985, p. 3). Participatory theater, made with, by and for the community, focused on process and without the necessity of public performance, gives space for stories to emerge organically, if and when that is healing; it obviates the aesthetic dilemma of who tells and who acts. But ethical challenges remain.

## 5. Levinas and Ethical Relations with the "Other"

The ethical concerns that emerge when working with vulnerable others require close attention. What marks out unbridgeable difference? How to serve others, without dis-

placing their dignity? How do we support the gradual, partial, recuperation of another person's humanity, or for that matter, one's own, particularly when power positions are so radically different? Wherein lies my responsibility? I am inadequate; yet, I am required to act. But *act*, how? In the face of compelling social and political needs, I hope to offer tools, 'expertise' and effort, and to do so with humility, whether through participatory work or through "writing on behalf of" as in plays such as *Testimonies*, my 2019 play about Yezidi survivors.

These questions are grounded in Emmanuel Levinas's "radical concept of ethics as unconditional responsibility". For Levinas, the face of the Other is the first absolute, the requisite on which the framework of ethical response is built. The ego is commanded by a transcendent order to take responsibility for the other person. Since the Other looks at me, I am responsible for him (Levinas cited in Balfour 2012, p. 202; Levinas 1985, p. 96).

Levinas insists on the "absolute alterity of the Other" (Drabinski 2013, p. 103). We are not the same; to annihilate difference is anti-ethical. By the "Other", Levinas meant the other human being, who is other not because he or she is different from me but because he or she is absolutely other and cannot be reduced to an identity. When we make categories, we erase the individual, which is the antithesis of ethical responsibility to the other.

The Other has needs that must be addressed: "To recognize the Other is to recognize a hunger. To recognize the Other is to give" (Levinas 1969, p. 75), where giving is without expectation of return, so as not to be simply exchange. This means that one cannot approach the Other "with empty hands and closed home" (1969, p. 172). It is not that I believe myself as under an obligation to give. I cannot not give, which is to say there is no gift. I have already been dispossessed by the face of the Other.

"It is Levinas's idea of the *active, responsive, corporal encounter* with alterity that is so pertinent to ethical responsibility" (Balfour 2012, p. 210, my italics). What Levinas calls for can be seen in moments of performance as an embodied, affective and interactive mode (p. 241). The "saying" for Levinas is corporal (embodied), active and responsive, engaged and always in process. This embodied encounter is at the core of theatrical expression.

With the concept of "saying", Levinas evokes "interactive human discourse" rather than the more purely representation "said". "Saying" is always provisional, subject to exegesis (as Levinas indicates in his discussions of Torah, there is a vast gulf between vigorous open-ended interpretation and the view that Torah is settled law). The "said" always reduces the other to the known, and controllable, the Same (Balfour 2012, p. 203, citing Crichley). In these terms, whereas scripted drama is the realm of the "said", participatory theater is a space for "saying", and the "Other" is less likely to be rendered and reduced to the "Same".

Theater is grounded in active, intimate, hands-on participation and personal connection: making theater with people from "other" backgrounds is very much, as Dwight Conquergood notes, about "knowing how", and "knowing who" (Conquergood 2002, p. 146).

## 6. Yezidi Oral Culture and Kurdish Theater

In an interview in 2021, Zêdan Xalaf, translator and researcher, spoke about the Shingal Lives project at the Kashkul Center for Arts and Culture at (AUIS) (Jango 2021). Xalaf, originally from Shingal (Sinjar), is a survivor of the 2014 Yezidi genocide and manages a team that collects oral histories from Yezidi elders, storytellers, and others.

> We have never written anything down in our own language in at least the last 500 years. Under Ottoman rule we were not allowed to because they were in power, and we as the Yezidi community were demonized by them. . . . the Ottomans burnt everything that belonged to the Yezidis and launched genocidal campaigns. Yezidis kept their oral tradition because . . . . you need to tell stories to survive, you need to make sense of your existence. The archive that we have in the Yezidi community is intangible; it's just in our memory. This is how we kept our history, through oral tradition, and how we still manage to keep our

tradition alive. They can't destroy it because it is in our minds. If you survive, those stories will survive with you, if you don't, then nothing survives.

More broadly, Kurdish theater draws on collective memory, traditional stories, music, and poetry to strengthen social cohesion and pride in Kurdish culture, to cultivate a sense of national unity, and to create an identity based on shared culture and language. From 1974 to 1991, Kurdish theater developed primarily in response to Baathist efforts to eliminate Kurdish nationalism.[31] Kurdish language was banned and replaced by Arabic in Iraqi schools. Kurdish plays were subjected to strict censorship nation-wide.[32]

In the 1980s, mainstream Kurdish theater turned to stylistic (non-political) experimentation. A 'Theatre of Images' developed in Erbil and Duhok, which foregrounded color, image, sound, and imaginative explorations. But in the mountains and villages, a form of guerrilla theater emerged. During the Iran–Iraq War (1980–1988), many Iraqi Kurds fled to the mountains, where cultural activities included theatrical performances aimed at political education. A prominent technique was Living Newspaper, which was developed in the Soviet Union and adopted in the US by the Federal Theater Project, in which news headlines were acted out for the benefit of largely illiterate audiences, to introduce and foster dialogue about controversial social issues. The style was heavily influenced by Brechtian epic theater, using simple sets, props, and costumes, to create socially conscious work that wrestles with political concerns. Later, some Iraqi Kurdish theater incorporated the work of Augusto Boal, as mentioned above.

Theater became a space for collective mourning after the Anfal.[33] In 1991, when thousands again fled to the border regions between Iran, Iraq and Turkey, performances began to take place in refugee and IDP camps. By the mid-1990s, theater groups emerge in the Autonomous Region of Iraq (KRI/Kurdish Region of Iraq) particularly in the cities of Erbil and Duhok, with the persistent goal of making Kurdish identity visible, and resisting cultural erasure. Memory is knowledge; memory is strength.

## 7. Me-T—The Middle East Theater Project

Theater is a social practice; in zones of conflict and contestation, theater can be a means to strengthen community, mediate conflict, and support dialogue across difference. Middle East Theater (ME-T) is an initiative provide space and opportunity for young people from across ethnic divides to enhance their skills and lead their communities. The initiative, begun in 2020 at the University of Duhok, focuses on research and building platforms for dialogue.

One ME-T project is titled: The Voice of ISIS War Survivors: A Platform for Youth Dialogue in Iraq, and it aims at using theater to explore the difficult past and build bridges toward a workable future. By sharing research and utilizing theater to generate dialogue, young Yezidi, Arabs, Kurds, and Christians build relationships, explore common aspirations, and consider competing claims and narratives. Principle strategies include dialogue, discussion, and social theater (particularly Boal's Forum Theater); story swaps, oral histories, field interviews; archival work and documentation; and building a digital archive of transgenerational memories across groups.

Dialogue across difference and between antagonists is crucial. Groups become trapped in their own stories, which then feed into cycles of reciprocal violence. Dialogue (may) facilitate reconciliation, and using tools of theater to air controversial issues, explore the roots and branches of conflict and analyze potential steps forward through participatory models such as Boal's Forum Theater can help to identify our differences, define goals and build bridges. In asymmetric conflicts, focus on commonalities is problematic if used by the dominant group to silence or diminish disavow the experiences of the minority. There is no need to agree, but there is a need to hear.

Clearly, in any political conflict, material conditions must change. As Antonio Gramsci warns, refashioning culture is not sufficient for producing social change; economic and political structures must also be transformed. ME-T works in the cultural as well as the political realm, with a focus on transforming and transcending habitual enmities, building

relationships and committing to reciprocal respect. By exposing each other to different versions of the same historical events, both commonalities and differences across groups can be productively explored. Large-scale problems are more likely to be resolved through cooperative efforts. The goal is to humanize the other, to fight xenophobic myths, and to determine factors (and challenges) that can be the basis for a superordinate identity. Making things together, focusing on cooperation and common aspirations, is the way forward. ME-T offers a platform for facilitating this work.

### 8. Re-Making Home

The Jewish definition of sin is something that is *out of place*. Refugee bodies, such as those of the internally *displaced* are out of place, through no fault of their own; they are reluctant residents, wherever they are. Political currents, material exigencies, and stereotyping mean refugees and IDPs are often treated and punished as sinners. But this formulation begs the question: Who belongs? Much of the work of SOHF and other theater projects helps participants to re-imagine home and to establish new ways of belonging. *Making* oneself at home is a continual and open-ended process of making the world and the self in mutuality (Arnold and Meskimmon 2015, p. 263).

SOHF is building a home for people that have lived for eight years in tents who have lost family, tribe and home. Everyone is welcome; those fleeing Mt. Sinjar and unstable encampments in Syria still arrive. Lisa Miara explains that once she had to remove ISIS "missionaries": "two young men, youth, came to us as rescues, but they tried to surreptitiously recruit for ISIS". But all who work (as staff) or study at SOHF are victims of ISIS or have renounced jihad. Miara says of them, "we love them to life".

The camp residents have been deeply traumatized, but they embrace hope. Material conditions have improved, and aspirations are fully voiced. A young musician, guided through his studies, is enrolled in the Fine Arts program at the University in Duhok. Hiba Qassim, a young woman from Sinjar, grew up in an IDP camp. She achieved the highest high school GPA in Iraq in 2020 and now is studying medicine on full scholarship, at UKH. Services such as SOHF offer affirmation, dignity, and, to some extent, opportunity.

"Daesh wanted to cover the whole world in black", they tell us. Young women who were raped, tortured, and dehumanized now wear pink, orange, green, red, every color, except black.

**Funding:** This research received no external funding.

**Institutional Review Board Statement:** Not applicable.

**Informed Consent Statement:** Not applicable.

**Data Availability Statement:** Not applicable.

**Conflicts of Interest:** The author declares no conflict of interest.

### Notes

[1]  "Saddam Hussein systematically destroyed Yezidi villages, then he collected our families together in one complex. In 1991 the name of the village was changed to Shariya". Sahla Eales (personal interview, 12 May 2022).

[2]  Duhok, a city of 180,000, houses over 400,000 refugees and IDPs. https://www.unrefugees.org/emergencies/iraq/ (accessed on 22 August 2022).

[3]  "IDPS are defined by the UN as persons or groups of persons who have been forced or obliged to flee or to leave their homes or places of habitual residence, in particular as a result of or in order to avoid the effects of armed conflict, situations of generalized violence, violations of human rights or natural or human-made disasters, and who have not crossed an internationally recognized state border". (Jeffers 2011, p. 17).

[4]  *Zamir* is Arabic for Diamond. Days before the mare gave birth, we were told that she would have twins, but there was just this baby; the other would have been named Gold.

[5]  Estimates of Yezidi population vary between half a million post-genocide (in *Yezidi Identity Politics in the Wake of the ISIS Attack*) to up to two million (in Khattar), but after the 2014 genocide, an accurate count is impossible.

6    The 1951 Refugee Convention defines refugee as "[a] person with a well-founded fear of being persecuted for reasons of race, religion, nationality, membership of a particular social group or political opinion, is outside the country of his nationality and is unable or, owing to such fear, is unable to avail himself of the protection of that country". (cited in UNHCR 2010).

7    By mid-2022, UNHCR Global Trends (2021) estimates that there are 84 million forcibly displaced people worldwide. https://www.unhcr.org/en-us/figures-at-a-glance.html (accessed on 22 August 2022).

8    Yezidism is considered one of the most ancient Eastern religions, and its followers believe that their religions originated from the ancient Babylonian religion that appeared thousands of years ago in Mesopotamia, and it is one of the religions that graduated from natural worship to monotheism and has its own beliefs and rituals that differ from the Abrahamic religions. https://whc.unesco.org/en/tentativelists/6467/ (accessed on 22 August 2022).

9    Among the many resources on Yezidi culture, I found the Khattar, S., Rostami, M. and Spat, E. cited below to be helpful.

10    Prior to 2014, there were some positive connections between neighbors. For example, before relations deteriorated, Yezidi and Muslim men in the region practiced *kreef*, a blood bond in which men participate in the circumcision of each other's sons and become co-equal godfathers.

11    According to the UN, 250,000 Yezidi from Sinjar were displaced in KRI, between 1500 and 5000 civilians were murdered; 5000 to 9000 were captured; as of June 2016, 2500 Yezidi women and children had escaped captivity or were ransomed as of June 2016; 1000s are missing or still in captivity.

12    As of May 2022, Turkish troops and Iraqi army forces have surrounded Mt. Sinjar, and the agreement with Yezidi self-defense militias have broken down. Turkish airstrikes recently hit Mount Sinjar, just as more than 150 Yezidi families had returned after living in IDP camps. Turkey justifies both the occupation of northern Syria and the airstrikes in Iraq as necessary to target PKK militants. But Turkish military operations also deter Yezidi civilians from returning to their lands.

13    Personal interview, March 2019.

14    The Kurds are a stateless people who have endured cultural genocide and disenfranchisement in Iraq, Syria, Turkey and Iran. Rojava (Autonomous Administration of North East Syria) is an attempt to carve out a space that is safe from ISIS, Turkish incursion, and the depredations of the Syrian civil war. it is best described as a noble but fragile experiment. Interviews with women, families, activists, artists, refugees and fighters tell a complicated story.

15    My larger project is to write a play about women's lived experience in Rojava, this autonomous region where female leadership is foregrounded. I plan to conduct discussions and interviews with the women of Kongreya Star, and Mala Jin, both women's education and social services organizations that deal with domestic violence and women's empowerment. I plan to spend time with families, talk with women soldiers, mothers, daughters, students, political actors, attend classes and workshops at the Universities of Kobani and Qamislo and visit Jinwar Free Women's Village, an ecological village presently under construction.

16    Raqaa is a city in Syria where many captives were taken and sold, but more broadly, it refers to towns and cities in Syria, Iraq, Saudi Arabia and other places where women were enslaved and children forced to join ISIS.

17    Mozarref Sheife is an Iranian–Kurdish actor and husband of ***Bayan Sami Abdul Rahman***, the Kurdistan Regional Government (KRG) Representative to the United States.

18    *Testimonies* had a public presentation (2021) at Silverthorne Theatre, Greenfield MA.

19    Prentki, Tim; Preston, Sheila (eds.). *The Applied Theatre Reader*. doi:10.4324/9780203891315 is a central resource for defining and exploring Applied Theater.

20    Augusto Boal's books *Theatre of the Oppressed, Games for Actors and Non-Actors, The Rainbow of Desire*, among dozens of other publications by and about Boal, offer a vital framework for using theater in non-theater settings, to explore issues of political and social importance.

21    In *Otherwise than Being or Beyond Essence* (Levinas 1991, Alphonso Lingis trans.), (Dordrecht: Kluwer Academic Publishers, 1978) Levinas considers the linguistic tension between the "Saying" and the "Said". The "Saying" is fluid, open to possibility, and as such able to respond to "the inconceivable nature of otherness", as Melissa Rachel Schwartz says in her 2017 dissertation, The Language of Ethical Encounter: Levinas, Otherness, & Contemporary Poetry (Schwartz 2017, p. 4) The "Said" is static, frozen, complete in itself. For further reading, see the *Two Aspects of Language: The Saying and the Said in The Intrigue of Ethics: A Reading of the Idea of Discourse in the Thought of Emmanuel Levinas*, Jeffrey Dudiak, Fordham, 2001, and Bernhard Waldenfels's "Levinas on the Saying and the Said", pp. 86–97 in *Addressing Levinas* (Nelson 2005).

22    For an extended discussion of Gilligan's theory of radical listening and ethical care, see Perspect Med Educ. 2017 Apr; 6(2): 76–81. Carol Gilligan and Jessica Eddy. Published online 2017 Mar 27. doi:10.1007/s40037-017-0335-3.

23    Personal interview, 16 May 2022.

24    Speaking at the College Women's Research Center in Amherst, MA (2000), Egyptian choreographer Nora Amin shared her experience working with British and Arab women who did not speak each other's language. Amin asked the women to embody their idea of "woman". They found commonality in physical portraits of bending, crouching, stooping over, weighed down by work. I utilized this technique with male and female groups, as we explored how gender plays into their lives. This led to hilarious skits, which was followed by discussion of women's roles and rights in Yezidi society.

[25]    The story is repeated in many versions of Kurdish tales; I used *A Fire in My Heart: Kurdish Tales* retold by Diane Edgecomb, Westport, CT, 2008, pgs. 95–96, as a principal source for the Kawa tale.

[26]    These include *Testimonies* (2019), noted above; *Someone is Sure to Come* (Kaplan 2019) based on work with Death Row inmates, presented at La Mama Annex, NYC and other venues in Massachusetts, published in Tacenda Journal; and plays that have been produced and performed but not published, including *Sarajevo Phoenix*; *Homeland/Homeless*, among others.

[27]    Jeffers, in *Refugees, Theatre, and Crisis*, discusses *CMI (A Certain Maritime Incident)* an Australian performance that rejects "notions of empathy and looks for an alternative way of engaging the natal Australian audience with questions of responsibility" (64–65). "Refusing to stage the bodies of refugees and show instead the obfuscations and evasions of the performative speech at the government enquiry represented a deliberately provocative challenge to the Australian audience asking, "Who speaks for whom, under what privilege and with what force?" (Dwyer quoted in Williams 2017, p. 202). "There were no refugees to be pitied and no refugee stories to sadden or enrage an audience. Instead audiences were confronted with . . . stories of the professionalism with which high ranking military men side-stepped and evaded the difficult question of how an untrue story had gained such a hold in the national rhetoric of asylum. They rejected the emphasis on placing the citizen in a position of empathy and ask instead that audiences take on a level of responsibility *as citizens*. (Jeffers 2011, p. 65) In *Refugee Performance*, Chapter 11: 'Politics Begins as Ethics': Levinasian Ethics and Australian Performance Concerning Refugees, Burvill addresses many of these issues.

[28]    See Parr (2021) and others.

[29]    For discussion of outsider as researcher and relative positions of power, see Hume and Mulcock (2004).

[30]    A theater piece featuring stories of refugees, first performed by the Théâtre du Soleil in 2003.

[31]    Second Iraqi–Kurdish War was led by Iraqi forces against rebel KDP troops of Mustafa Barzani during 1974–1975. The war came in the aftermath of the First Iraqi–Kurdish War (1961–1970), as the 1970 peace plan for Kurdish autonomy had failed to be implemented by 1974 (Wikipedia, Iraq-Kurdish conflict).

[32]    I am indebted to Mari Rostami's book-length study of Kurdish Theater for this overview.

[33]    The Anfal refers to Saddam Hussein's attacks on Kurds, most notably in Halabja, in 1988. Overall, 182,000 Kurds died, some 5000 in chemical attacks orchestrated by the notorious Chemical Ali.

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
