# Peer review of "Refuge and Resistance: Theater with Kurds and Yezidi Survivors of ISIS"

_humanities, doi:10.3390/h11050111_

Round 1
Reviewer 1 Report
It is with great pleasure that I recommend this outstanding article for publication. Drawing upon the author’s extensive experience as a researcher and a theatre practitioner, the article analyzes the efficacy and intricacies of applied theatre practice within the Yezidi community of the Shariya Camp for Displaced Persons in Kurdistan, a semi-autonomous region in northern Iraq. The article begins with a detailed description of the Yezidi community and its place within a broader narrative of Kurdish history. In this way the article situates its study within the complexities of the region’s cultural and religious history, most importantly in reference to the Yezidi genocide in 2014 at the hands of ISIS. This historical overview offers essential cultural context to readers who are not specialists in the region and lends depth and detail to the author’s subsequent discussion of the nuances of the applied theatre practice at the centre of the article’s analysis. By locating its study so specifically within the history and culture of the Yezidi people, the article offers important insight into the nature of applied theatre practice with vulnerable communities more broadly. In this way, the author makes an essential contribution to a broader discourse about socially engaged theatre internationally, and also sets an elegant precedent for how an in-depth ethnographic approach to analysis yields remarkably insightful and original research.
Of particular interest in this regard is the author’s reflection on the nature of insiders and outsiders in this type of research. Despite the author’s clear “cultural competency,” as they put it, they still acknowledge their own presence in the production process as an outsider. And yet, the author reports, it is in part this outsider-ness that offers validation to the participants who have experienced an historical invisibility or lack of recognition on the international stage. This is an area of inquiry that could be expanded upon, it seems to me, either in this article or elsewhere, as it resonates far beyond the specifics of this study and highlights important questions about the nature of ethnographic research more broadly. Questions about how linguistic and cultural competency come to shape one’s work as a theatre researcher, particularly in the context of working with vulnerable communities, seem to me a somewhat understudied topic that the author of this article is uniquely placed to address.
On this note, I was especially impressed by the article’s clear and nuanced analysis on the difference between doing theatre about vulnerable groups vs doing theatre with vulnerable groups. Drawing important distinctions between the politics of verbatim theatre and those of applied theatre, the author points directly to many of the pressing ethical concerns at the heart of contemporary applied theatre practice. Building in particular on the work of Emmanuel Levinas, the article makes an especially original and insightful observation in distinguishing between participatory theatre as a practice of “saying,” i.e. a reciprocal process of speaking and hearing, in distinction to something that has been “said” which, according to the author, reduces the narrative to something that is known, and controllable. This distinction between the “saying” and the “said” and its alignment with textual vs participatory performance practice, carries with it a depth of analysis that is applicable to studies far beyond the regional and cultural specificity with which it is applied in this article.
These are among the major contributions this article makes to the fields of theatre studies, refugee studies, and to the development of ethnography as a methodological approach that by its nature must come under constant scrutiny as it adjusts to each historical and cultural context to which it is applied. I look forward to seeing this article in publication and to following the author’s subsequent work on this topic and others.
Author Response
Thank you!
Reviewer 2 Report
This article is excellent, and most certainly suitable for publishing in that special issue of Humanities. Its topic is relevant, and it is as sensitive as one would wish in such painful situations.
It opens with a lively description of a theatrical event, and then proceeds to sketch the historical background of the traumatized people. Rhetorically this works very well. The description is gripping and the history horrible. The special features of the Yezidi people, such as their hybrid religion, predict the violence they will encounter at the hands of ISIS. The author then presents their concept of “applied theatre” as a socially engaged way of helping to create a public space of the healing of collective trauma. The historical and traditional stories, rituals, and more are reworked to enliven the present. The key of this theory of theatre is to work with the people on stories about them.
For the special issues on refugees and representation this “working through” and “working with” is very relevant. It avoids the realistic bias that would bring representations about refugees dangerously close to the kind of voyeurism Adorno was already so agitated about after WWII. The artistic work with theatre is eminently apt to make that avoidance concrete and yet, not shy away from the people’s own concerns, traditions and histories.
The article is very well written and accessible to everyone. It is well-documented and instructive, not only for those interested in theater but for everyone who cares about the connections between art and social-political life - which is the most important mission of the Humanities.
My only tiny suggestion is to eliminate the full stop after headings, which is done after heading 5 and 8.
Author Response
Thank you!